# Stability Studies of 16 Antibiotics for Continuous Infusion in Intensive Care Units and for Performing Outpatient Parenteral Antimicrobial Therapy

**DOI:** 10.3390/antibiotics11040458

**Published:** 2022-03-29

**Authors:** Guillaume Loeuille, Elise D’Huart, Jean Vigneron, Yann-Eric Nisse, Benoit Beiler, Caroline Polo, Gillian Ayari, Matthieu Sacrez, Béatrice Demoré, Alexandre Charmillon

**Affiliations:** 1Pharmacy Department, University Hospital, 54511 Vandoeuvre-lès-Nancy, France; g.loeuille@chru-nancy.fr (G.L.); j.vigneron@chru-nancy.fr (J.V.); yann-eric.nisse@ch-remiremont.fr (Y.-E.N.); b.beiler@chru-nancy.fr (B.B.); c.polo@chru-nancy.fr (C.P.); g.ayari@chru-nancy.fr (G.A.); m.sacrez@chru-nancy.fr (M.S.); b.demore@chru-nancy.fr (B.D.); 2Infostab, Non-Profit Association, 54180 Heillecourt, France; 3EA 4360 APEMAC, Lorraine University, 54000 Nancy, France; 4Infectious Diseases Department in Charge of Mobile Infectiology Team University Hospital, 54511 Vandoeuvre-lès-Nancy, France; a.charmillon@chru-nancy.fr

**Keywords:** stability, antibiotic, intensive care unit, outpatient parenteral antimicrobial therapy, continuous infusion

## Abstract

The use of continuous infusion to improve the therapeutic efficacy of time-dependent antibiotics has been demonstrated. There is still a lack of data to safely perform these continuous infusions. The objectives in this study were to evaluate the stability by using stability-indicating methods (High-Performance Liquid Chromatography) of 16 antibiotics in concentrated solutions, especially for administration in intensive care units and solutions in elastomeric diffusers at 37 °C for outpatient parenteral antimicrobial therapy. The solutions were considered stable if the percentage of the drug was ≥90%, and the colour and clearness remained unchanged. In syringes, the stability data vary from 4 to 8 h (h) for meropenem in Dextrose 5% (D5W) and Normal Saline (NS), respectively, 6 h for cefotaxime, 12 h for cefoxitin, and 24 h for aztreonam, cefazolin, cefepime, cefiderocol, ceftazidime/avibactam, ceftolozane/tazobactam in NS and D5W, and in water for injection for cloxacillin. A stability period of 48 h has been validated for vancomycin (D5W), aztreonam, and piperacillin/tazobactam. Cefoxitin, cefazolin, cefepime, cefotaxime, cloxacillin, and piperacillin are unstable for diffuser administration. In diffusers, stability times vary from 6 h for cefiderocol, 8 h for ceftazidime, 12 h for ceftazidime/avibactam and ceftolozane/tazobactam (NS), 24 h for temocillin (NS) and piperacillin/tazobactam (D5W), up to 48 h for aztreonam and vancomycin. Solutions stored at 37 °C are less stable and allow the administration of seven antibiotics using diffusers.

## 1. Introduction

The value of continuous infusion for time-dependent antibiotics has been demonstrated for several years. In particular, for beta-lactam antibiotics, therapeutic optimisation via continuous infusion makes it possible to target a free plasma concentration of beta-lactam antibiotics between 4 and 8 times the Minimum Inhibitory Concentration (MIC) of the incriminated bacteria, and thus to obtain a maximum rate of bactericidia [1]. A meta-analysis including 13 randomised controlled trials evaluating the impact of continuous or discontinuous beta-lactam administration in critical care patients showed improved clinical recovery in septic patients and patients at high risk of mortality, as well as in patients treated with continuous infusion [2]. The use of continuous infusion is also recommended in cases of high MIC of the causative bacteria or critical care patients with non-fermenting Gram-negative bacilli (*A. baumanii* and *P. aeruginosa*) to improve the clinical cure rate [3,4]. For vancomycin, a meta-analysis demonstrated that continuous infusion compared to intermittent administration is associated with a 53% reduction in the risk of acute kidney injury without influencing overall mortality [5].

The use of antibiotic infusions in critical care is frequent. A large European multicentre study reported the use of antibiotics in 64% of patients during their hospitalisation in an intensive care unit (ICU) [6]. In these ICUs, under real-life conditions, antibiotic infusions are often carried out in small volumes at high concentrations. The question then arises of the stability of these high-concentration solutions over a long period of time to achieve these continuous infusions.

Upon discharge from the hospital, patients can continue treatment with Outpatient Parenteral Antimicrobial Therapy (OPAT). The major benefits of OPAT are the reduction or avoidance of hospital stays, the reduction in nosocomial infections and hospital-related conditions, significant cost savings, and improved quality of life for the patient [7,8]. The use of continuous home infusion may be recommended for infections where the risk of failure is high, for infections with high MIC bacteria, or for infections where the delivery of beta-lactams is difficult, such as foreign-material infections [9,10,11]. Portable elastomeric devices allow these continuous infusions to be performed in the patient’s home. Information provided by pharmaceutical companies in the Summaries of Products Characteristics (SmPC) does not provide details for continuous intravenous administration of highly concentrated solutions for administration in syringes in ICUs or for solutions at an elevated temperature (32 or 37 °C) in elastomeric devices for continuous infusion at home (OPAT) [12].

It is, therefore, crucial to carry out stability studies to safely perform these continuous infusions. As specified in the ASEAN guidelines (2005), the stability of an active ingredient or a pharmaceutical speciality is its ability to maintain its properties within specific limits throughout its lifetime. The chemical, physical, microbiological, and biopharmaceutical aspects of stability must be considered [13]. Drug solutions are usually considered chemically stable if they retain their chemical characteristics: The percentage of the drug intact remains above 90% of the initial concentration [14]. The stabilities of injectable antibiotics have been published, but often at too-low concentrations, which do not allow a small volume for the preparation of syringes for continuous administration over 12 or 24 h or have not been studied at body temperature for administration at home using elastomeric diffusers [15].

The aim of this study was to determine the physicochemical stability of 16 antibiotics, namely amoxicillin, aztreonam, cefazolin, cefepime, cefiderocol, cefotaxime, cefoxitin, ceftazidime, ceftazidime/avibactam, ceftolozane/tazobactam, cloxacillin, meropenem, piperacillin, piperacillin/tazobactam, temocillin, and vancomycin, under two conditions for continuous infusion. The first condition is in a high-dose polypropylene syringe at 20–25 °C for ICUs. The second condition is at 37 °C in portable elastomeric devices for OPAT. Only amoxicillin was packaged in polyolefin bags for administration using a volumetric pump and stored at room temperature [16].

To study the physico-chemical stability of a drug, a determination of the active ingredient must be carried out using a separative analytical method, in order to be able to separate the active substance from its degradation products. To perform these stability studies, we used High-Performance Liquid Chromatography (HPLC) [17].

## 2. Results

### 2.1. Chemical Stability by HPLC

The pre-study made it possible to eliminate certain antibiotics due to the appearance of a precipitate after storage at 37 °C (cefazolin, cloxacillin) or a change in the colour of the solution (cefotaxime, cefoxitin, and piperacillin).

The validation method criteria for HPLC analysis are presented in Table 1. Linearity was demonstrated for all molecules, with an R^2^ value between [0.9981 and 0.9999]. The calibration curves agreed with the linear model via a non-linearity test ANOVA (Fexp < Fth = 3.71) for all molecules. The homogeneity of the variances was also proven by a Cochran test (Cexp < Cth = 0.684) for all molecules. Intraday and interday precisions were less than 2.5%. The analytical methods were stability-indicating.

Table 2 presents the concentrations of antibiotics in polypropylene syringes and Table 3 presents the concentrations in elastomeric containers. The concentrations of amoxicillin obtained in polyolefin bags are presented in Table 2. Several examples of chromatograms obtained after reconstitution (T0 h) are presented in Figure 1.

### 2.2. The pH Measurement

pH value variations were less than one pH unit except for 125 mg/mL cefazolin in both solvents in syringes (6 g in 48 mL) after 48 h, 125 mg/mL cefoxitin in NS after 12 h, and piperacillin/tazobactam 16 g/0.75 g in NS in 240 mL in an elastomeric container after 24 h.

### 2.3. The Visual and Subvisual Evaluation

During the stability study, several physical changes were observed, with a major intensification of yellowing for solutions whose chemical stability specifications were maintained. After 12 h, cefotaxime solutions at 83.3 and 125 mg/mL in syringes and ceftolozane/tazobactam at 25/12.5 mg/mL in diffusers after 24 h presented visual modifications. These physical changes were observed in both NS and D5W.

The particle counter test was not available at the beginning of the stability studies and was carried out on new syringes at the end of the studies by measuring the number of particles inferior to 10 µm and 25 µm according to European Pharmacopoeia. This test was only performed on antibiotics that were visually and chemically stable. All samples were within the specifications of European Pharmacopoeia.

## 3. Discussion

### 3.1. Choice of the Temperature

When administered at home, the elastomeric container is placed under the clothes near the body. The temperature of the solution is higher than the classical ambient temperature at 25 °C and is near body temperature. Some authors or guidelines suggest using 32 °C for these stability studies considering that the device is not in the body and that the temperature is lower [18]. Other teams work by using body temperature, considering the worst condition that can be met in some hot countries depending on geographical area. In our studies, we decided to use the latter conditions.

### 3.2. Choice of Molecules and Study Design

Certain time-dependent antibiotics were not studied in our stability studies, such as ceftriaxone or ertapenem, because of their long half-life requiring only one injection per day. Conversely, molecules with well-defined instability, such as imipenem/cilastatin (stable for 4 h at a concentration of 5 mg/mL), were not studied [19], as well as meropenem in a diffuser. The study of temocillin in syringes was not performed as the laboratory had already provided these stability data in NS and D5W [20]. Amoxicillin was studied at 20 mg/mL in polyolefin bags to validate data of the literature review of Diamantis et al. [17]. Stability up to 12 h has been demonstrated, allowing two infusions per day by a volumetric pump (4 g/200 mL or 6 g/300 mL, for example).

### 3.3. About the Solvents Used

NS and D5W were used for all the products except cloxacillin, where SWFI was used. Stability can be different according to the solvent, depending on the pH value (D5W has a pH around 4 and NS around 6–7) or the chloride ions present in NS. Some products are not stable in both solvents. The choice of the solvent should be respected in accordance with stability studies. For the solution of cloxacillin at 250 mg/mL, the solvent used must be SWFI (12 g in 48 mL) and not NS or D5W due to precipitation. The osmolality of the solution is approximately 550 mOsmol/L, which allows intravenous administration [21]. We were able to validate the reconstitution of cloxacillin powder with only 3 mL of SWFI, allowing only one syringe of 12 g of cloxacillin to be made in 48 mL.

### 3.4. Limiting Factors

Due to poor solubility of piperacillin/tazobactam parenteral injection powder, higher concentrations could not be studied in syringes, as 12 g in 48 mL requires extensive shaking and ultrasonication to ensure complete dissolution, which was not feasible in ICUs.

According to the manufacturer’s recommendations, a 1 g vial of cefiderocol should be reconstituted with 10 mL of solvent, which limits us to making a 48 mL syringe. Due to its high cost, the reduction of the reconstitution volume was not studied, which made it impossible to manufacture a 125 mg/mL (6 g/48 mL) syringe [22].

### 3.5. Citrated Buffered Solutions

An interesting approach to enhance the stability of very unstable drugs is to buffer the solution near pH 7 by using a citrated buffered solution. British teams used infusion bags of 0.3% sodium citrate in NS to study the stability of meropenem, piperacillin/tazobactam, and flucloxacillin. For 10 and 50 mg/mL flucloxacillin solutions, they demonstrated stability for 13 days stored at 2–8 °C followed by 24 h at 32 °C in two elastomeric containers (Accufuser and INfusor LV) [15]. In comparison, unbuffered flucloxacillin solutions lost up to 60% after storage at 37 °C for 24 h [23].

For piperacillin/tazobactam, extended stability was demonstrated with up to 13 days 2–8 °C plus 24 h at 32 °C “in-use” when using FOLFusor LV10 (Baxter) or Easypump II (B. Braun) pump devices [18]. However, these results were not observed for meropenem at concentrations between 6.25 mg/mL and 25 mg/mL, which was not sufficiently stable to administer as a 24-h infusion in ambulatory device reservoirs [24].

An Australian team studied the stability of benzylpenicillin and flucloxacillin after reconstitution with 4% sodium citrate and dilution in NS infusion bags. This approach was chosen because citrated infusion bags were not available on the Australian market.

Benzylpenicillin (15 and 60 mg/mL) and flucloxacillin (5 and 60 mg/mL) infusions in LV Elastomeric Infusor devices and 0.9% sodium chloride Viaflex bags were prepared as buffered and unbuffered solutions. Buffering was achieved by reconstituting antibiotic vials with sodium citrate 4%. Infusions were stored at 2–8 °C for 6 days then 37 °C for 24 h. Buffered benzylpenicillin 15 and 60 mg/mL and flucloxacillin 5 and 60 mg/mL in LV Elastomeric Infusors and 0.9% sodium chloride Viaflex bags appear chemically stable for 6 days refrigerated, as well as for a subsequent 24 h at 37 °C. Unbuffered solutions, prepared in NS, presented high instability. After 6 days at 2–8 °C and 1 day at 37 °C the concentration of 60 mg/mL benzylpenicillin in LV Infusor was 42.9% of the initial concentration instead of 100.3% for the buffered solution. Under the same storage conditions, the percentages of the initial concentration for 60 mg/mL flucloxacillin solutions were 63.1 and 99.6 [25]. These results showed the great potential of using citrated buffered solutions for molecules unstable in NS or D5W. However, this approach should be validated by a stability study for each molecule. 

This is, to our knowledge, the first study to summarize the stability of 16 antibiotics, evaluated with the same robust methodology. It aimed to provide clinicians with a practical document to refer to when they want to optimize their treatment on a PK/PD level or to facilitate OPAT.

## 4. Materials and Methods

### 4.1. Chemical, Reagents and Products Used

The products used for the preparation of the mobile phase and the validation of the analytical method were of HPLC grade. Water for chromatography was obtained from a reverse-osmosis system (Millipore Iberica, Madrid, Spain), with a resistivity <15 MΩ cm. The antibiotic drugs used for the preparation of the tested solutions are summarized in Table 4. Normal saline (NS), sterile water for injection (SWFI), and 5% dextrose (D5W) used for the reconstitution of vials or the dilution of drugs were purchased from polyolefin bags (Easyflex, Macopharma, France) or glass vials (Chaix et du Marais, Lavoisier, France). For the preparation of solution tests, drugs were stored in polypropylene syringes (BD Plastipak 50 mL), in polyisoprene elastomeric devices (Baxter FOLFusor 5 mL/h or 10 mL/h), or in polyolefin bags (amoxicillin: Easyflex, Macopharma, Mouvaux, France).

### 4.2. Apparatuses

The following apparatuses were used for the stability studies:–The High-Performance Liquid Chromatography (HPLC) system consisted of an ELITE LaChromVWR/ Hitachi plus autosampler, a VWR photodiode array detector L- 2455, and a VWR L-2130 HPLC pump. Data were acquired and integrated using EZChrom Elite (VWR, Agilent).–pH meter (Bioblock Scientific model 93313).–PAMAS particle counter, Rutesheim, Germany.

### 4.3. Methods

Considering the preparation of test solutions and storage, the choice of concentrations, solvents, and analysis times was based on a collegial decision between an infectious disease specialist and a pharmacist in relation to observed practices. Syringe preparations were performed at qs 48 mL and qs 120 or 240 mL in elastomeric devices. Each preparation corresponded to the total daily dose to be administered. The preparations of solutions tested are presented in Table 5.

Syringes and polyolefin bags (amoxicillin) were stored at room temperature, without protection from light and elastomeric devices at 37 °C in a climatic chamber. The stabilities of the antibiotics were studied at different analysis times over 48 h. For each condition, three syringes/bags or three elastomeric devices were prepared, and three samples for each preparation were chemically analysed.

To avoid time-consuming chemical stability studies by HPLC, a pre-study in glass vials was carried out at 37 °C at the concentrations used in elastomeric containers. The objective was to evaluate the physical stability by searching for the formation of precipitate or a change in colour to eliminate unstable drugs. If a physical modification was observed, this was concluded as physical instability, and the study of the chemical stability in diffusers was not carried out.

### 4.4. Chemical Stability by HPLC

Antibiotics solutions were analysed by stability-indicating reversed-phase HPLC methods adapted from previous publications.

The analytical conditions used for HPLC analysis (composition of the mobile phase, pH, flow rate, injection volume, wavelength for detection, retention time, and reference of the publication used for the choice of the method) are presented in Table 6.

Separation was performed on a LichroCART Merck C18 analytical column (5 µm, 125 mm × 4.0 mm) for all methods except ceftazidime, where a LichroCART Merck C18 column (5 µm, 250 mm × 5.0 mm) was used.

For four analytical methods (cefiderocol, cefotaxime, ceftazidime/avibactam, and piperacillin/tazobactam), a gradient elution mode was used.

Chemical stability was defined as no less than 90% of the initial concentration in relation to the evolution of potential degradation products [14].

### 4.5. Validation of the Analytical Methods

The validation of analytical methods was performed as recommended by the International Conference on Harmonisation Q2R1 (ICH) [40]. The calibration curves were constructed from the plots of peak area versus concentration. The linearity of each method was evaluated with 5 concentrations. For all calibration curves, the homogeneity of the variances was evaluated with a Cochran test of which the significance level was set at *p* < 0.05. An analysis of variance (ANOVA) of the linear regression data was performed to assess the significance (*p* < 0.05) of the proposed methods. Intra-day reproducibility and inter-day precision were evaluated. The limit of detection (LOD) was determined graphically for each analytical method. Selectivity and specificity were evaluated by forced degradation studies. For each analytical method, acid, basic, thermic, photolytic, and oxidative degradation were performed. The objective was to obtain a degradation between 10 and 20% of our molecules of interest to prove our methods are stability-indicating [17].

### 4.6. Physical Stability

Physical stability was defined as the absence of particulate, haze formation, or a colour change. The samples were visually inspected against a white/black background with the unaided eye at each analysis timepoint. Physical stability was also assessed by performing a particulate contamination test (PAMAS SVSS) at the beginning and the end of the study. The results were analysed according to the criteria of the European Pharmacopoeia [41,42].

### 4.7. pH Measurements

pH was measured at each time of analysis. A variation of more than one pH unit was considered unacceptable.

### 4.8. Summary of the Results

A summary of the results, including the physical and chemical stability usable in daily practice, is presented in Table 7.

## 5. Conclusions

Among the sixteen antibiotics tested at elevated concentrations, one was stable in polypropylene syringes for only 4 h (meropenem in D5W), one for 6 h (cefotaxime in D5W and NS), two for 8 h (ceftazidime (D5W) and meropenem (NS)), one for 12 h (cefoxitin in D5W), seven for 24 h (cefazolin, cefepime, cefiderocol, ceftazidime/avibactam in NS and D5W, ceftazidime and piperacillin in NS, cloxacillin (SWFI) at 250 mg/mL and 125 mg/mL in NS and D5W), and five were stable for 48 h (aztreonam, ceftolozane/tazobactam, piperacillin/tazobactam in NS and D5W, piperacillin and vancomycin in D5W). In elastomeric containers stored at 37 °C, we demonstrated 6 h stability for cefiderocol (NS and D5W), 8 h stability for ceftazidime (NS and D5W), ceftolozane/tazobactam (D5W), and piperacillin/tazobactam (in NS), 12 h stability for ceftolozane/tazobactam and ceftazidime/avibactam in NS, 24 h stability for piperacillin/tazobactam in D5W and temocillin (NS), and 48 h stability for aztreonam and vancomycin (NS and D5W). Amoxicillin (NS) in a polyolefin bag was stable for 12 h.

Clinicians must be aware that stability in syringes does not mean stability in diffusers. The choice of solvent must also be respected in accordance with these stability studies due to notable differences between NS, D5W, and SWFI. The concentration of the antibiotics studied must also be respected. It is possible to modify the dose/volume. Amoxicillin stability has been demonstrated at 20 mg/mL in bags for 12 h at room temperature. These new data allow us to perform administration for 12 h at 2 g/100 mL or at 4 g/200 mL, 5 g/250 mL, or 6 g/300 mL, for example. Regarding the diffusers, solutions must be protected from light by being worn under the patient’s clothes, for example, or by a suitable device.

To our knowledge, this is the first study to summarize the stability data of 16 antibiotics at dosages relevant for use in clinical practice, evaluated with the same robust methodology. These new stability data allow for improved drug therapy and safer administration of these continuous infusions. The intention is to provide clinicians with a practical document to refer to when they want to optimize their treatment in terms of PK/PD or to facilitate OPAT.

## Figures and Tables

**Figure 1 antibiotics-11-00458-f001:**
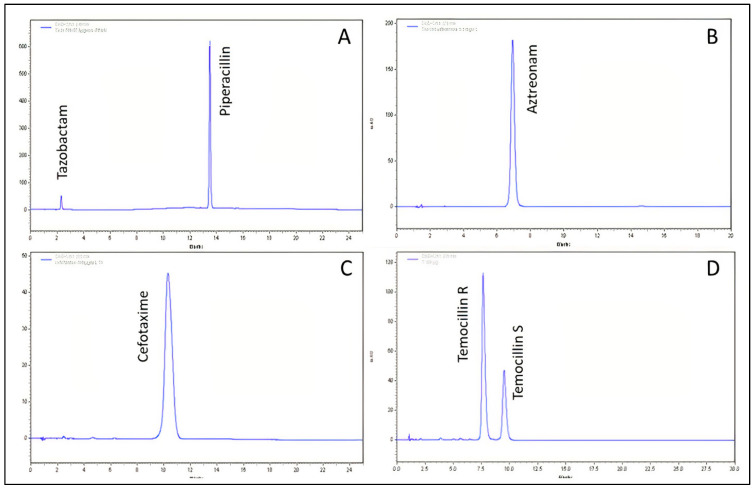
Examples of chromatograms of piperacillin/tazobactam (300/37.5 µg/mL) (**A**); aztreonam (100 µg/mL) (**B**); cefotaxime (100 µg/mL) (**C**); and temocillin R and S (150 µg/mL) (**D**), obtained immediately after reconstitution in Normal Saline solution.

**Table 1 antibiotics-11-00458-t001:** Validation criteria for analytical HPLC methods.

Antibiotic	Calibration Range (µg/mL)	R^2^	Intra-Day Precision [min; max] (%)	Inter-Day Precision [min; max] (%)	Limit of Detection [µg/mL]	Cochran’s Test C_exp_	ANOVA (Non-Linearity) F_exp_
Amoxicillin	120–280	0.9999	[0.06; 0.66]	[0.57–0.75]	0.13	0.344	0.15
Aztreonam	50–150	0.9997	[0.08; 1.48]	[1.18; 1.95]	0.51	0.399	0.93
Cefazolin	75–175	0.9999	[0.15; 0.85]	[0.44; 0.57]	0.11	0.609	0.38
Cefepime	60–140	0.9999	[0.04; 0.83]	[1.15; 1.70]	0.36	0.269	0.13
Cefiderocol	25–75	0.9999	[0.05; 1.53]	[0.41; 0.99]	0.10	0.424	0.07
Cefotaxime	50–150	0.9998	[0.08; 1.81]	[1.09; 1.66]	2.02	0.600	1.93
Cefoxitin	75–175	0.9993	[0.17; 2.04]	[1.40; 1.80]	0.53	0.420	0.66
Ceftazidime	100–500	0.9999	[0.02; 1.53]	[0.28; 0.94]	0.23	0.533	3.06
Ceftazidime/Avibactam	100–500	0.9999	[0.24; 0.50]	[0.31; 1.05]	2.43	0.551	0.21
	25–125	0.9999	[0.13; 0.43]	[0.33; 0.72]	3.17	0.514	1.38
Ceftozolane/Tazobactam	50–250	0.9999	[0.07; 1.93]	[0.62; 1.47]	0.77	0.579	0.26
	25–125	0.9999	[0.06; 2.04]	[0.63; 1.60]	0.84	0.643	0.49
Cloxacillin	1200–2800	0.9981	[0.33; 1.81]	[1.25; 1;95]	4.26	0.575	0.92
Meropenem	50–250	0.9999	[0.06; 1.30]	[0.71; 1.13]	0.19	0.659	0.64
Piperacillin	100–300	0.9999	[0.01; 0.46]	[0.55; 1.10]	0.34	0.555	0.06
Piperacillin/Tazobactam	100–500	0.9999	[0.12; 1.33]	[0.38; 1.62]	0.23	0.599	0.99
	12.5–62.5	0.9999	[0.01; 1.28]	[0.76; 1.44]	0.36	0.554	0.14
Temocillin	50–250	0.9999	[0.05; 1.72]	[0.91; 2.02]	0.64	0.331	0.04
Vancomycin	50–150	0.9995	[0.03; 1.65]	[1.70; 2.48]	2.91	0.389	2.27

**Table 2 antibiotics-11-00458-t002:** Stability of antibiotics in polypropylene syringes and amoxicillin in polyolefin bags at 20–25 °C for continuous intravenous infusion measured by HPLC.

			Mean % of Initial Concentration ± RSD * %						
Antibiotic	Conc.	Solvent	T0 h	T4 h	T6 h	T8 h	T12 h	T24 h	T48 h
Amoxicillin **	20 mg/mL	NS	100 ± 1.19	-	96.7 ± 1.77	-	94.1 ± 0.66	87.5 ± 0.45	78.5 ± 1.52
Aztreonam	125 mg/mL	NS ***	100.0 ± 2.38	-	98.7 ± 0.90	-	-	95.3 ± 1.79	92.7 ± 1.95
		D5W ****	100.0 ± 0.47	-	99.5 ± 0.77	-	-	98.1 ± 1.42	95.7 ± 0.57
Cefazolin	125 mg/mL	NS	100.0 ± 2.35	-	103.3 ± 1.67	-	-	100.0 ± 1.98	100.6 ± 1.95
		D5W	100.0 ± 0.91	-	99.8 ± 3.11	-	-	99.4 ± 1.99	97.7 ± 2.10
Cefepime	110 mg/mL	NS	100.0 ± 2.15	-	98.1 ± 0.61	-	-	94.8 ± 1.85	87.3 ± 0.82
		D5W	100.0 ± 1.78	-	94.7 ± 0.93	-	-	90.6 ± 0.43	85.5 ± 1.80
Cefiderocol	62.5 mg/mL	NS	100.0 ± 1.30	-	-	-	94.2 ± 1.11	91.6 ± 0.98	85.7 ± 2.24
		D5W	100.0 ± 1.43	-	-	-	97.4 ± 0.68	94.0 ± 0.73	87.1 ± 0.53
Cefotaxime	83.3 mg/mL	NS	100.0 ± 2.15	-	95.3 ± 1.50	-	92.2 ± 1.93	87.2 ± 2.05	-
		D5W	100.0 ± 1.48	-	95.4 ± 1.04	-	93.0 ± 1.43	88.3 ± 1.45	-
	125 mg/mL	NS	100.0 ± 1.70	-	95.7 ± 1.38	-	95.2 ± 1.23	51.4 ± 1.53	-
		D5W	100.0 ± 1.21	-	97.4 ± 2.00	-	94.9 ± 1.18	-	-
Cefoxitin	125 mg/mL	NS	100.0 ± 1.88	-	99.5 ± 0.79	-	97.3 ± 3.73	94.4 ± 1.60	89.8 ± 2.03
		D5W	100.0 ± 1.79	-	98.9 ± 1.32	-	96.4 ± 3.00	93.8 ± 1.80	89.6 ± 2.14
Ceftazidime	125 mg/mL	NS	100.0 ± 0.94	-	-	98.3 ± 2.04	-	94.9 ± 1.80	86.5 ± 1.89
		D5W	100.0 ± 1.69	-	-	95.1 ± 1.06	-	89.0 ± 0.78	82.2 ± 0.44
Ceftazidime	125 mg/mL	NS	100.0 ± 0.98	-	-	-	96.7 ± 0.31	95.2 ± 1.33	87.4 ± 1.22
		D5W	100.0 ± 1.26	-	-	-	98.1 ± 0.86	91.7 ± 0.55	87.1 ± 1.36
Avibactam	31.25 mg/mL	NS	100.0 ± 0.86	-	-	-	96.4 ± 0.91	96.2 ± 1.77	91.0 ± 0.89
		D5W	100.0 ± 1.47	-	-	-	99.1 ± 0.79	94.7 ± 0.77	93.0 ± 1.41
Ceftozolane	62.5 mg/mL	NS	100.0 ± 2.15	-	-	96.4 ± 2.25	-	93.9 ± 2.25	91.8 ± 1.52
		D5W	100.0 ± 1.92	-	-	98.5 ± 1.55	-	96.2 ± 2.50	92.8 ± 0.54
Tazobactam	31.25 mg/mL	NS	100.0 ± 2.27	-	-	98.0 ± 2.57	-	99.0 ± 2.30	101.0 ± 2.34
		D5W	100.0 ± 2.09	-	-	100.0 ± 1.79	-	101.3 ± 2.74	102.3 ± 0.83
Cloxacillin	250 mg/mL	SWFI *****	100.0 ± 1.91	-	98.8 ± 1.70	-	-	96.2 ± 1.63	90.2 ± 1.45
	125 mg/mL	NS	100.0 ± 3.07	-	100.2 ± 1.93	-	-	97.2 ± 3.01	90.5 ± 2.73
		D5W	100.0 ± 1.48	-	100.0 ± 2.06	-	-	97.3 ± 1.47	90.1 ± 1.21
Meropenem	41.7 mg/mL	NS	100.0 ± 1.59	97.1 ± 0.66	-	93.0 ± 0.93	-	-	-
		D5W	100.0 ± 1.52	93.8 ± 0.68	-	85.9 ± 0.74	-	-	-
Piperacillin	125 mg/mL	NS	100.0 ± 0.87	-	97.6 ± 0.24	-	-	92.8 ± 2.05	88.9 ± 1.37
		D5W	100.0 ± 1.11	-	100.0 ± 1.97	-	-	98.3 ± 1.23	97.5 ± 1.26
Piperacillin	125 mg/mL	NS	100.0 ± 0.96	-	-	101.3 ± 0.44	-	101 ± 1.26	98.1 ± 1.45
		D5W	100.0 ± 2.34	-	-	97.1 ± 0.59	-	95.9 ± 1.05	93.5 ± 0.68
Tazobactam	15.6 mg/mL	NS	100.0 ± 1.16	-	-	100.8 ± 0.40	-	101.1 ± 1.56	99.4 ± 1.52
		D5W	100.0 ± 2.26	-	-	96.3 ± 0.36	-	96.7 ± 1.49	95.1 ± 1.22
Temocillin	Unrealized in syringe								
Vancomycin	62.5 mg/mL	NS	100.0 ± 1.65	-	99.8 ± 1.17	-	-	100.7 ± 1.05	99.5 ± 1.27
		D5W	100.0 ± 0.50	-	99.3 ± 1.03	-	-	98.2 ± 1.34	94.6 ± 2.88
	83.3 mg/mL	NS	100.0 ± 1.84	-	99.4 ± 1.28	-	-	98.4 ± 2.06	-
		D5W	100.0 ± 1.62	-	100.8 ± 0.92	-	-	96.0 ± 6.31	101.0 ± 0.86

* RSD: Relative standard deviation; ** packaged in polyolefin bag; *** NS: Normal Saline, **** D5W: Dextrose 5%, ***** Sterile Water for Injection.

**Table 3 antibiotics-11-00458-t003:** Stability of antibiotics in polyisoprene elastomeric devices for use in OPAT at 37 °C measured by HPLC.

			Mean % of Initial Concentration ± RSD *					
Antibiotic	Conc.	Solvent NS/D5W	T0 h	T6 h	T8 h	T12 h	T24 h	T48 h
Aztreonam	50 mg/mL	NS **	100.0 ± 1.97	-	-	-	102.5 ± 3.39	100.2 ± 2.17
		D5W ***	100.0 ± 1.64	-	-	-	101.7 ± 1.67	95.3 ± 2.96
Cefazolin	50 mg/mL	NS	Unstable, precipitate formation during pre-study					
		D5W						
Cefepim	50 mg/mL	NS	100.0 ± 1.43	93.2 ± 2.00	-	-	83.3 ± 2.28	59.5 ± 2.21
Cefiderocol	Unrealized in elastomeric device							
Cefotaxime	25 mg/mL	NS	Unstable, colour change after 6 h during pre-study					
		D5W						
Cefoxitine	25 mg/mL	NS	Unstable, Colour change after 12 h during pre-study					
		D5W						
Ceftazidime	25 mg/mL	NS	100.0 ± 3.22	-	95.2 ± 1.56	-	85.6 ± 2.24	-
		D5W	100.0 ± 2.94	-	94.8 ± 3.29	-	77.5 ± 1.94	-
Ceftazidime	25 mg/mL	NS	100.0 ± 3.35	-	-	92.2 ± 2.86	82.3 ± 2.28	66.3 ± 1.90
		D5W	100.0 ± 7.35	-	-	86.2 ± 6.54	75.4 ± 7.53	58.1 ± 8.56
Avibactam	6.25 mg/mL	NS	100.0 ± 4.59	-	-	98.4 ± 4.08	96.5 ± 2.95	93.9 ± 2.95
		D5W	100.0 ± 7.81	-	-	95.4 ± 6.92	93.9 ± 7.46	90.3 ± 8.35
Ceftozolane	25 mg/mL	NS	100.0 ± 1.80	-	100.0 ± 1.20	-	91.8 ± 0.95	81.4 ± 3.67
		D5W	100.0 ± 0.99	-	97.5 ± 0.85	-	89.3 ± 2.11	81.7 ± 6.02
Tazobactam	12.5 mg/mL	NS	100.0 ± 2.08	-	104.6 ± 1.47	-	109.2 ± 1.33	116.7 ± 3.02
		D5W	100.0 ± 0.99	-	103.1 ± 0.82	-	109.5 ± 1.33	116.6 ± 1.83
Cloxacillin	50–100 mg/mL	NS	Unstable, precipitate formation during pre-study					
		D5W						
Meropenem	Unrealized in elastomeric device							
Piperacillin	66.7 mg/mL	NS	Unstable, precipitate formation during pre-study					
		D5W						
Piperacilline	66.7 mg/mL	NS	100.0 ± 2.17	-	98.5 ± 0.46	-	93.6 ± 1.06	85.7 ± 1.92
		D5W	100.0 ± 0.74	-	97.9 ± 0.87	-	93.6 ± 0.60	84.1 ± 0.17
Tazobactam	8.3 mg/mL	NS	100.0 ± 2.26	-	98.9 ± 0.53	-	97.0 ± 0.93	96.2 ± 1.58
		D5W	100.0 ± 0.78	-	98.7 ± 0.90	-	98.2 ± 0.55	95.8 ± 0.25
Temocillin	25 mg/mL	NS	100.0 ± 2.37	-	-	-	92.6 ± 2.95	80.4 ± 2.84
		D5W	100.0 ± 2.36	-	-	-	87.5 ± 2.23	78.8 ± 2.61
Vancomycin	37.5 mg/mL	NS	100.0 ± 2.06	-	-	-	97.9 ± 2.91	98.3 ± 3.26
		D5W	100.0 ± 2.70	-	-	-	101.0 ± 1.61	103.3 ± 1.54

* RSD: Relative standard deviation; ** NS: Normal Saline, *** D5W: Dextrose 5%.

**Table 4 antibiotics-11-00458-t004:** List of antibiotics drugs used for the preparation of solutions.

	Tradename/Manufacturer	Batch Number
Amoxicillin	Amoxicilline PANPHARMA 1 g	307197
Aztreonam	AZACTAM^®^ 1 g SANOFI-AVENTIS	ABC7060
Cefazolin	Céfazoline MYLAN 2 g	200902–200903
Cefepime	Céfépime MYLAN 2 g	4M2119FR
Cefiderocol	FETCROJA^®^ 1 g SHIONOGI	FEFR0120
Cefotaxime	Céfotaxime MYLAN 2 g	R3052
Cefoxitin	Céfoxitine PANPHARMA 2 g	N4-03
Ceftazidime	Ceftazidime MYLAN 2 g	191102
Ceftazidime/Avibactam	ZAVICEFTA^®^ 2/0.5 g PFIZER	3M05L95690
Ceftozolane/Tazobactam	ZERBAXA^®^ 1/0.5 g MERCK SHARP & DOHME BV	T003341
Cloxacillin	ORBENINE^®^ 1 g ASTELLAS	25AND02/ 25AQF03
Meropenem	Méropénem PANPHARMA 1 g	MFR1020
Piperacillin	Pipéracilline PANPHARMA 4 g Pipéracilline PANPHARMA 1 g	306609–306699 306421
Piperacillin/Tazobactam	Pipéracilline/tazobactam PANPHARMA 4/0.5 g	306584
Temocillin	NEGABAN^®^ 1 g EUMEDICA NEGABAN^®^ 2 g	L154510 L162439
Vancomycin	Vancomycine SANDOZ 1 g	EC0107

**Table 5 antibiotics-11-00458-t005:** Preparation of test solutions.

Antibiotic	Polyolefin Bag (100 mL, 20–25 °C)		Syringe (48 mL, 20–25 °C)		Elastomeric Device (37 °C)	
	Amount (g)	Solvent	Amount (g)	Solvent	Amount (g)	Solvent
	(Concentration mg/mL)		(Concentration mg/mL)		(Concentration mg/mL)	
Amoxicillin	2 g 20 mg/mL	NS *	Unrealized			
Aztreonam	Unrealized		6 g (125 mg/mL)	NS—D5W **	6 g (120 mL) (50 mg/mL)	NS—D5W
Cefazolin	Unrealized		6 g (125 mg/mL)	NS—D5W	6 g (120 mL) (50 mg/mL)	NS—D5W
Cefepime	Unrealized		6 g (125 mg/mL)	NS—D5W	6 g (120 mL) (50 mg/mL)	NS
Cefiderocol	Unrealized		3 g (62.5 mg/mL)	NS—D5W	6 g (240 mL) (25 mg/mL)	NS—D5W
Cefotaxime	Unrealized		4 g (83.3 mg/mL)	NS—D5W	6 g (240 mL) (25 mg/mL)	NS—D5W
			6 g (125 mg/mL)	NS—D5W		
Cefoxitin	Unrealized		6 g (125 mg/mL)	NS—D5W	6 g (240 mL) (25 mg/mL)	NS—D5W
Ceftazidime	Unrealized		6 g (125 mg/mL)	NS—D5W	3 g (120 mL) (25 mg/mL)	NS—D5W
Ceftazidime/Avibactam	Unrealized		6/1.5 g (125/31.25 mg/mL)	NS—D5W	3/0.75 g (120 mL) (25/6.25 mg/mL)	NS- D5W
Ceftozolane/Tazobactam	Unrealized		3/1.5 g (62.5/31.25 mg/mL)	NS—D5W	3/1.5 g (120 mL) (25/12.5 mg/mL)	NS—D5W
Cloxacillin	Unrealized		12 g (250 mg/mL)	SWFI ***	12 g (120 mL) (100 mg/mL) 12 g (240 mL) (50 mg/mL)	NS—D5W
			6 g (125 mg/mL)	NS—D5W		
Meropenem	Unrealized		2 g (41.7 mg/mL)	NS—D5W	Unrealized	
Piperacillin	Unrealized		6 g (125 mg/mL)	NS—D5W	16 g (240 mL) (66.7 mg/mL)	NS—D5W
Piperacillin/Tazobactam	Unrealized		6/0.75 g (125/15.6 mg/mL)	NS—D5W	16 g/2 (240 mL) (66.7/8.3 mg/mL)	NS—D5W
Temocillin	Unrealized		Unrealized ^1^		6 g (240 mL) (25 mg/mL)	NS—D5W
Vancomycin	Unrealized		3 g (62.5 mg/mL)	NS—D5W	4.5 g (120 mL) (37.5 mg/mL)	NS—D5W
	Unrealized		4 g (83.3 mg/mL)			

* NS: Normal Saline, ** D5W: Dextrose 5%, *** Sterile Water for Injection. ^1^ stability data available in the SmPC of temocillin.

**Table 6 antibiotics-11-00458-t006:** List of HPLC conditions for the antibiotic stability studies.

Antibiotic	Mobile Phase (*v*/*v*)	pH	Flow Rate (mL/min)	Injection Volume (µL)	Wavelength (nm)	Retention Time (min)	Reference
Amoxicillin	Isocratic: NaH_2_PO_4_ buffer 0.05 M/methanol (95/5)	4.4	1.0	50	220	4.45	[26]
Aztreonam	Isocratic: KH_2_PO_4_ buffer 0.05 M/methanol (90/10)	3.0	1.0	20	270	6.9	[27]
Cefazolin	Isocratic: KH_2_PO_4_ buffer 0.005 M/methanol (80/20)	7.5	1.0	50	272	3.0	[28]
Cefepime	Isocratic: KH_2_PO_4_ buffer 0.005 M/methanol (90/10)	7.5	1.0	50	257	3.8	[28]
Cefiderocol	Gradient: KH_2_PO_4_ buffer 0.05 M (A) + methanol (B) T0 to T7 min gradual increase 83/17 (A/B) to 70/30; T7 to T15 min: 70/30; T16 to T20 min: 83/17	3.0	1.5	50	260	5.9	[29]
Cefotaxime	Gradient: Na_2_HPO_4_ buffer 0.05 M/methanol (86/14) (A); Na_2_HPO_4_ buffer 0.05 M/methanol (60/40) (B) T0 to T7 min: 100/0 (A/B); T9 to T16 min: 80/20; T16 to T30 min: gradual increase until 41.4/58.6; T35 to T40 min: 100/0	6.25	1.3	10	235	9.0	[30]
Cefoxitin	Isocratic: KH_2_PO_4_ buffer 0.005 M/methanol (80/20)	7.5	1.0	10	272	3.2	[28]
Ceftazidime	Isocratic: ammonium acetate 0.1 M/acetonitrile 90/10	7.5	1.0	20	260	4.1	[31]
Ceftazidime/Avibactam	Gradient: KH_2_PO_4_ buffer 0.05 M (A) + methanol (B) T0 to T4 min: 99/1 (A/B); T9 to T28 min: 90/10; T36 to T40 min: 99/1	3	1.5	20	260	20.3/1.6	[32]
Ceftozolane/Tazobactam	Isocratic: KH_2_PO_4_ buffer 0.005 M/acetonitrile (1000/26)	3.4	1.0	20	220	8.7/4.8	[33]
Cloxacillin	Isocratic: triethylamine + tetrabutylammonium buffer/methanol (35/65)	6	0.5	5	250	4.5	[34]
Meropenem	Isocratic: ammonium acetate 10.53 mM /acetonitrile (95/5)	3.0	1.0	20	297	8.1	[35]
Piperacillin	Isocratic: KH_2_PO_4_ buffer 0.05 M/acetonitrile (55/45)	3.0	1.0	2	230	3.4	[36]
Piperacillin/Tazobactam	Gradient: KH_2_PO_4_ 0.02 M (A)/acetonitrile (B) T0 to T5 min: 92.5/7.5 (A/B); T10 to T15 min: 70/30; T20 to T25 min: 92.5/7.5	2.5	1.5	10	210/280	13.5/	[37]
						2.2	
Temocillin	Isocratic: KH_2_PO_4_ buffer 0.1 M/methanol (93/7)	7.0	1.0	20	230	7.6 and 9.2	[38]
Vancomycin	Isocratic: KH_2_PO_4_ buffer 0.1 M/acetonitrile (92/8)	3.5	1.5	10	220	7.2	[39]

**Table 7 antibiotics-11-00458-t007:** Stability data for antibiotics in syringes, polyolefin bags *, or elastomeric devices.

Antibiotic	Syringe (48 mL, 25 °C), Polyolefin Bags * (100 mL, 25 °C)			Diffuser (37 °C)		
	Amount (g)	Solvent	Stability (Hours)	Amount (g)	Solvent	Stability (Hours)
	(Concentration)			(Concentration)		
Amoxicillin *	2 g (100 mL) (20 mg/mL)	NS **	12 h	Unrealized in elastomeric device		
Aztreonam	6 g	NS-D5W ***	48 h	6 g (120 mL) (50 mg/mL)	NS-D5W	48 h
	(125 mg/mL)					
Cefazolin	6 g	NS-D5W	24 h	6 g (120 mL) (50 mg/mL)	NS-D5W	Precipitate formation during the pre-study
	(125 mg/mL)					
Cefepime	6 g	NS-D5W	24 h	6 g (120 mL) (50 mg/mL)	NS	Visual modification after 6 h at 37 °C
	(125 mg/mL)					
Cefiderocol	3 g	NS-D5W	24 h	6 g (240 mL) (25 mg/mL)	NS- D5W	6 h
	(62.5 mg/mL)					
Cefotaxime	4 g–6 g	NS-D5W	6 h	6 g (240 mL) (25 mg/mL)	NS-D5W	Colour change after 6 h during the pre-study
	(83.3–125 mg/mL)					
Cefoxitin	6 g	D5W	12 h	6 g (240 mL) (25 mg/mL)	NS-D5W	Instability during the pre-study
	(125 mg/mL)					
Ceftazidime	6 g	NS	24 h	3 g (120 mL) (25 mg/mL)	NS-D5W	8 h
	(125 mg/mL)	D5W	8 h			
Ceftazidime/	6/1.5 g	NS-D5W	24 h	3/0.75 g (120 mL)	NS	12 h
Avibactam	(125/31.25 mg/mL)			(25/6.25 mg/mL)	D5W	Unstable
Ceftozolane/	3/1.5 g	NS-D5W	48 h	3/1.5 g (120 mL)	NS D5W	12 h 8 h
Tazobactam	(62.5/31.25 mg/mL)			(25/12.5 mg/mL)		
Cloxacillin	12 g	SWFI ****	24 h	6–12 g (120 mL) (50–100 mg/mL)	NS-D5W	Precipitate formation during the pre-study
	(250 mg/mL)					
	6 g	NS-D5W	24 h			
	(125 mg/mL)					
Meropenem	2 g	NS	8 h	Unrealized in elastomeric device		
	(41.7 mg/mL)	D5W	4 h			
Piperacillin	6 g	NS	24 h	16 g (240 mL) (66.7 mg/mL)	NS-D5W	Instability during the pre-study
	(125 mg/mL)	D5W	48 h			
Piperacillin /	6/0.75 g	NS-D5W	48 h	16/2 g (240 mL)	NS	8 h
Tazobactam	(125/15.6 mg/mL)			(66.7/8.3 mg/mL)	D5W	24 h
Temocillin	Unrealized			6 g (240 mL)	NS	24 h
				(25 mg/mL)	D5W	Unstable
Vancomycin	3 g	D5W	48 h	4,5 g (120 mL) (37.5 mg/mL)	NS-D5W	48 h
	(62.5 mg/mL)					
	4 g	D5W	48 h			
	(83.3 mg/mL)					

* packaged in polyolefin bag, ** NS: Normal Saline, *** D5W: Dextrose 5%, **** Sterile Water for Injection.

## Data Availability

Not applicable.

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
