# Peer review of "Stability Studies of 16 Antibiotics for Continuous Infusion in Intensive Care Units and for Performing Outpatient Parenteral Antimicrobial Therapy"

_antibiotics, 2022, doi:10.3390/antibiotics11040458_

Round 1
Reviewer 1 Report
The present manuscript entitled "Stability studies of 16 time-dependent antibiotics for continuous infusion in intensive care units and for performing outpatient parenteral antimicrobial therapy" by Guillaume Loeuille, Elise D'Huart, Jean Vigneron, Yanne-Eric Nisse, Benoit Beiler, Caroline Polo, Gillian Ayari, Matthieu Sacrez, Béatrice Demoré, Alexandre Charmillon (antibiotics-1652256) describes the studies on the physicochemical stability of 16 time-dependent antibiotics. The chemical stability was carried out with the use of high-performance liquid chromatography with a photodiode array detector. Additionally, an assessment of physical stability was also performed. Obtained data can be useful in improving drug therapy and infusing safety.
The present article is written correctly and has a good structure; moreover, it has all the necessary parts. The article is interesting from an analytical and medical point of view. The paper meets Antibiotics' requirements, and I recommend the article for publication in Antibiotics following the common editing stage. My current decision is a major revision. More specific comments and observations are presented below.
- Abstract. Information on the analytical technique used can be added to this section.
- Page 2, line 68. Typo error in “chemvvbnical”.
- Introduction. A paragraph on applied analytical techniques for this type of research may be added.
- Validation. LOD values should be calculated and placed in Table 1.
- How many and on what points were the calibration plots built?
- Exemplary chromatograms can be added.
- Has the interference been studied? What can be done in the event of strong interference effects? How would you deal with them? What types of interference effects could occur?
- Tables. You can write "- ± -" as "-".
- Page 8, line 149. [19] should be instead of (19).
- Page 9, line 198. What were the parameters of the water used?
- How were the temperature, time, light, and humidity parameters selected? Were you planning to use experimental planning methods?
- Was each substance analysed separately? There was no one method for all?
- Table 6. Please extend the description of the gradient.
- Table 7 should be placed in Discussion section.
- Conclusion. Please, emphasize clearly the advantages of the research carried out.
- References. Please adjust the references following the requirements of the journal.
I hope that the comments presented will help improve the article.
Reviewer 2 Report
The authors have used continuous infusion to improve the therapeutic efficacy of time-dependent antibiotics. They evaluated the stability of 16 time-dependent antibiotics in concentrated solutions, especially for administration in ICUs and solutions in elastomeric diffusers at 37°C for outpatient parenteral antimicrobial therapy. The study is thorough and well designed, however, there are a few concerns that should be addressed by the authors.
- Storage conditions can affect the stability of antibiotics infusions. What were the storage conditions used for the antibiotics?
- Authors analyzed the physical stability and chemical stability of the infusions. Did the authors confirm physical stability using microscopic imaging as well?
- Authors studied these stabilities at varied temperatures. The authors should explain if relative humidity will play any role here?
- Type “chemvvbnical” line number 68, the third paragraph of the introduction.
Author Response
First of all, we would like to thank the review committee of the journal for their pertinent comments on this work. If you agree, we changed the title and the main document without the term “time-dependent antibiotics” when we listed the number of antibiotics studied. Indeed, vancomycin is a time-dependent but also a concentration-dependant antibiotic. Using only the term "time dependent" when listing the antibiotics studied (including vanomycin) can lead to confusion.
Comments and suggestions for authors : 2
- The conditions of storage (temperature, protection of light, containers, duration of storage) are presented in line 238-241.
- For all stability studies performed in this work, a visual examination has been realised.
European Pharmacopoeia suggests two methods for the analysis of non visible particles : particle counter and microscope. We used a particle counter available in our laboratory. Microscope could be selected as complementary method, more usually used for the evaluation of suspensions or emulsions, and was not selected for our stability studies.
- The influence of humidity has not been evaluated in our studies. Indeed, the ICH guidelines recommend a residual humidity rate close to 60%. Our containers, polypropylene syringes and polyisoprene elastomer devices, are hermetic containers, so humidity has little influence on this type of administration. The impact of humidity is more crucial for solid oral forms as tablets or capsules.
- We have corrected this error.
Round 2
Reviewer 1 Report
Dear Authors,
Thank you for your meticulous consideration of my comments. The paper has improved substantially and, to my opinion, is suitable for publication.